# The Effects of the Progressive Replacement of Meat with Texturized Pea Protein in Low-Fat Frankfurters Made with Olive Oil

**DOI:** 10.3390/foods11070923

**Published:** 2022-03-23

**Authors:** Isabel Revilla, Sergio Santos, Miriam Hernández-Jiménez, Ana María Vivar-Quintana

**Affiliations:** Area of Food Technology, Polytechnic School of Zamora, Universidad de Salamanca, Avenida Requejo 33, 49022 Zamora, Spain; id00698408@usal.es (S.S.); miriamhj@usal.es (M.H.-J.); avivar@usal.es (A.M.V.-Q.)

**Keywords:** pea protein, backfat replacement, meat replacement, olive oil, fatty acid composition, texture, color, emulsion stability

## Abstract

There is growing interest in using healthy ingredients for the formulation of meat-based products. Among them, the replacement of pork fat with vegetable oils has attracted much attention. On the other hand, the use of vegetable proteins to replace meat provides multiple possibilities which have not been sufficiently studied. The aim of this study was to produce low-fat frankfurters in which all the pork fat had been replaced with olive oil and then to progressively replace (25%, 50%, 75% and 100%) the pork with textured pea protein. Texture, color, technological properties such as emulsion stability and cooking loss, proximate composition, and the fatty acid profile were analyzed. The results show that frankfurters made only with olive oil were slightly pale; however, they showed better emulsion stability and a healthier lipid profile than the 100%-meat-based frankfurters. Regarding the replacement of meat with texturized pea protein in the frankfurters made with olive oil, it was possible to replace up to 50% of the meat, and although significant differences were observed in terms of moisture, color, and texture, the product obtained showed similar values to other low-fat frankfurters.

## 1. Introduction

In recent years the meat industry has evolved rapidly and new products have appeared, owing to the need to reduce costs and new consumer trends towards healthier and more sustainable products [1]. Furthermore, vegetarian and vegan movements are increasing worldwide, which has stimulated the interest of the industry in plant-based products [2]. The reformulation of meat products into this type of products could fulfill the two main requirements of recent research: avoiding or reducing undesired substances and increasing the levels of healthy ingredients [3].

Although meat has played a decisive role in human evolution and in the current diet due to its nutritional characteristics, especially its content of high-quality protein, vitamins (B12 and D), and minerals (Fe and Se), several epidemiological studies have established that the consumption of processed meats is associated with an increased risk of cancers, cardiovascular disease, and diabetes [4]. These diseases are related to high fat consumption and the presence of carcinogens generated through various cooking and processing methods [5,6]. However, fat is an essential component of meat products because it affects technological properties such as emulsion stability in frankfurters [7] and sensory characteristics such as a salty taste [8] or texture [9].

One of the possibilities for reformulating meat products is the substitution of animal fat for vegetable fat in order to reduce the saturated fatty acid levels and to increase the level of unsaturated fatty acids by adding olive oil [7], grape seed oil [10], or soybean oil [11], because the influence of the amount and type of dietary fats on cancer has been demonstrated [12]. However, it should be noted that saturated fats contribute towards the texture, mouthfeel, juiciness, and sensory acceptability of products such as frankfurters [13,14]. This strategy is often accompanied by a reduction in total fat content to achieve a healthier product. Thus, Koutsopoulos et al. [15] made sausages with a 66% reduction in total fat, of which 20% was olive oil, with the addition of carrageenan. In their studies Choi et al. [16] reduced the amount of animal fat by up to two thirds, replacing it with up to 33% of olive, grape seed, corn, canola, and soybean oils and supplementing the formulation with rice bran fiber to obtain promising functional results in all of them. Jiménez-Colmenero [17] detailed different methods of incorporating non-animal fat either as solid fat, pre-emulsified oil, or liquid oil, whereas Lurueña-Martínez et al. [7] reduced the fat content by 40% and included up to 60% of the total fat as olive oil, but in no case has complete substitution been achieved. In all these studies the efficacy of hydrocolloids for the maintenance of structure has been demonstrated [18]. In particular, xanthan gum has been used in low-fat products owing to its water retention and texture improvement properties [19], and products have been developed in which no significant differences with the control product can be found [7].

On the other hand, the replacement of fat and meat with vegetable proteins is also being investigated due to their potential as extenders, their high nutritional value, and their wide range of functional properties such as solubility, viscosity, and water-holding capacity [20], which may be of interest regarding fat-free products, reducing undesirable sensations and improving mouthfeel and taste [21]. In addition, they can increase cooking yields and lower raw material costs [20].

Most studies have investigated the feasibility of replacing fat or meat with soy proteins, either in the form of flours or as dry or wet texturizates [22,23,24,25]. Good results have therefore been obtained in terms of physicochemical and textural properties using soy as a fat substitute in bologna-type meat emulsions and frankfurters combined with other hydrocolloids such as konjac gels or carrageenans [19,22,23] or with other compounds such as citrus fiber [26], chia flour, and inulin [27]. Regarding the substitution of meat for soy proteins, early studies used substitutions of up to 4% to obtain softer products [22]. More recent work points to the feasibility of replacing meat with texturized vegetable protein (TVP) from soy with good sensory acceptance for levels of up to 30% of total meat for beef sausages [24] and up to 80% in chicken sausages [25]. When the percentage of soy protein (concentrate or isolate) substitute for meat reaches 100%, lower hedonic ratings can be obtained for both flavor and texture than in products with meat [28]. In general, products made using soy proteins were characterized by lower hardness, higher WHC, and a yellower color [24,25,28] but also higher emulsion stability [25]. The use of other pulses as meat substitutes has recently been extensively reviewed [1] and these authors found that their application in cooked products has been much less studied. The results indicate that chickpea is a suitable fat replacer in bologna-type products as the final product showed improved sensory and instrumental texture properties without foreign off-flavors, whereas pea flour was rated the lowest in texture and flavor characteristics, in addition to showing a darker color [29,30]. As for the use of legumes as meat replacers in cooked comminuted products, the few results available show that replacing meat with pea protein isolate resulted in a lower pH, L* value, hardness, flavor, and texture sensory scores and a higher water content than the German boiled sausage which achieved the highest scores [28].

In this context, the aim of this study was to examine the feasibility of the total substitution of olive oil for animal fat in the production of low-fat frankfurters and then to proceed to the progressive replacement of lean meat with texturized pea protein, in addition to determining its effect on the stability and quality of the frankfurters.

## 2. Materials and Methods

### 2.1. Ingredients

Fresh pork shoulder lean meat, backfat pork, spices (garlic, onion, and black pepper) and olive oil (0.4° Carbonell, Cordoba, Spain. Fatty acid composition: C16:0 11.9%, C18:0 3.8%, C18:1 71.6%, C18:1n7 2.4%, C18:2n6 6.6%), were purchased in a local market (Zamora, Spain) and kept refrigerated at 4 °C until use. Texturized pea protein was supplied by the Dacsa Group (Valencia, Spain).

The other ingredients used were as follows: phosphates (E 451i, E 450i; Proanda, Seville, Spain), sodium chloride, sodium nitrite (E-250, Merck, Darmstadt, Germany), sodium lactate (E 325, Panreac, Barcelona, Spain); sodium ascorbate (E 301, Panreac, Barcelona, Spain); dextrose (D (+) Glucose, Merck Eurolab, Briare Le Canal, France); walnut tree smoke (AFS Mega 2000, Amcan, Le Chesnay, France); potato flour (Proanda S. A., Seville, Spain); soy protein (Proanda S. A., Seville, Spain), locust bean/xanthan gum commercial mix (Premigum XME-54, Premium Ingredients S. L., Murcia, Spain), and NF4166A and NF4167A flavorings (Lucta, Barcelona, Spain).

### 2.2. Production of Frankfurters

Six different formulations (Table 1) of about 3 kg each were prepared. Two trials of each preparation were performed on different days and with different meat and backfat.

The Meat 40%OO formulation was prepared by replacing 40% of the backfat with olive oil. This formulation showed no significant differences from a low-fat frankfurter elaborated only with backfat regarding emulsion stability, texture, or consumer acceptability [7]. Meat 100%OO frankfurter was prepared by replacing all the backfat with olive oil. The remainder of the formulations were prepared by gradually replacing the meat of Meat 100%OO formulation with 25%, 50%, 75%, or 100% texturized pea protein.

The frankfurters were prepared at the pilot plant of the Food Technology area. The soy protein was added to a bowl cutter (Talsa T-2473) equipped with three blades with a third of the ice at low speed (1500 r.p.m). Afterwards, lean meat or texturized pea protein was added with a further third of the ice, phosphates, and nitrite salt (99.4% sodium chloride with 0.6% sodium nitrite) at high speed (3000 r.p.m) to facilitate protein solubility. Subsequently, the backfat (Meat 40%OO) or olive oil was added with the remaining third of the ice at high speed until a homogeneous mass was formed. Finally, the remainder of the solid ingredients (sodium lactate, sodium ascorbate, dextrose, walnut tree smoke, potato flour, locust bean/xanthan gum spices, and flavorings) was added at low speed. The final temperature of the emulsion was lower than 10 °C.

Immediately after chopping, the batter was mixed under a vacuum (60 cm Hg) for 3 min [7]. The batter was then stuffed using a piston stuffer (Talsa H262A) into 22 mm diameter cellulose casings (Viscofan, Pamplona, Spain). Frankfurters were hand-linked at 15 cm intervals, weighed, and heat-processed in an Eller oven (Unimatic Micro model, Eller, Merano, Italy) according to the following processing cycle: drying for 15 min at 55 °C and 60% relative humidity (RH); heating for 15 min at 60 °C and 75% RH; and steam cooking at 75 °C to an internal temperature of 72 °C, monitored throughout using thermocouples inserted in the thermal center of the frankfurters. The frankfurters were then showered to a temperature of 20 °C and chilled at 2 °C overnight. After chilling, the frankfurters were weighed, peeled, and vacuum-packed (Tecnotrip V220) in special pouches for cooking (Tecnotrip 95MY, outside polyamide, inside polyethylene), then pasteurized in hot water at 75 °C for 45 min, and subsequently cooled in cold running water for 15 min before being stored in the dark at 4 °C for subsequent analysis.

### 2.3. Emulsion Stability

The measure of emulsion stability was carried out according to the following method [31]. Twenty-five grams (exact weight recorded) of the raw emulsion was placed in a centrifuge tube (five replicates per formulation) and centrifuged for 1 min at 2958 g using a Sigma 4K15 centrifuge (Osterode am Harz, Germany). The samples were heated in a water bath for 30 min at 70 °C and then centrifuged again for 3 min at 2958 g. The pelleted samples were removed and weighed and the supernatants were poured into pre-weighed crucibles and dried overnight at 100 °C. The volume of the total expressible fluid (%TEF) and the fat percentage were calculated as follows:TEF = (Weight of centrifuge tube and sample) − (Weight of centrifuge tube and pellet),(1)
% TEF = TEF/sample weight × 100(2)
EFat% = ((Weight of crucible + dried supernatant) − (Weight of empty crucible)/TEF) × 100(3)

### 2.4. Cooking Loss

Cooking loss values were determined by calculating the weight difference of five links of frankfurters before and after cooking and expressing the losses as a percentage of the initial weight.

### 2.5. Texture Analysis

The texture characteristics of emulsions, cooked frankfurters, and texturized pea protein were analyzed using a TA-XT2i texturometer (Stable Micro Systems, UK).

The texture of emulsions was determined using the back-extrusion procedure [32]. To do so, 100 g of emulsion samples obtained after the comminution process were carefully scooped into a back-extrusion cylindrical container (50 mm of internal diameter) and tempered until reaching 20 °C. A 40 mm compression disc was used and one compression cycle was applied at a constant speed of 1 mm/s to 20% of sample height, and the maximum back-extrusion force was recorded. The analyses were performed in triplicate.

The texture properties of the texturized pea protein were analyzed according to the method proposed by Osen et al. [33]. A square-shaped sample (27 × 27 mm) was cut longitudinally (longitudinal strength, FL) and in parallel (transverse strength, FT) in the direction of the fibers of the texturized protein using a knife blade. The samples were cut to 75% of their original thickness at a speed of 2 mm/s and the cutting strength was recorded. All determinations were performed with at least 12 replicates.

The texture of the cooked frankfurters was determined by means of a texture profile analysis (TPA) as described by Ordoñez, Rovira and Jaime [34]. A 50 mm cylindrical cell was used to compress the frankfurter samples (diameter 2.2 cm; height 1 cm) to 50% of their original height at a speed of 1 mm/s to determine the hardness (g), springiness (mm), cohesiveness (dimensionless), gumminess (g), fracturability (g), adhesiveness (g × mm), and chewiness (g × mm). The samples were heated in advance in a water bath (15 min, 70 °C). Ten replicates of each sample were carried out.

### 2.6. Proximate Composition

The chemical analysis of the frankfurters (moisture, protein, and fat and ash content) was assessed according to the AOAC methods [35]. Moisture was determined via oven-drying (AOAC 950.46), total fat via the Soxhlet method using ethylic ether (AOAC 985.15), ash via incineration at 550 °C (AOAC 920.153), and total protein via the Kjeldahl method using 6.25 as a conversion factor (AOAC 992.15). All the analyses were performed in triplicate.

### 2.7. Color

The color was measured using a HunterLab MiniScan colorimeter of the XE Plus model (Hunterlab, Virginia, USA) equipped with a 25 mm measuring head and diffuse/8° optical geometry and the CIELab parameters determined were L* (lightness), a* (redness), and b* (yellowness) using an observer of 10° and the illuminant D_65_. In order to measure the color of the frankfurters, these were cut longitudinally and the color was measured on its internal surface. Ten replicates per formulation were performed.

The color of the olive oil was measured using 25 mL of oil in a 2.5 inch glass sample cup was were covered by a ceramic disk and the entire assembly was covered using the sample cup’s opaque cover. The measurements were performed in triplicate.

### 2.8. Fatty Acid Profile

Intramuscular lipids were extracted using the chloroform/methanol procedure described by Folch et al. [36]. The fatty acid composition of lipids was determined according to the method described by Lurueña-Martínez et al. [37]. Extracted fatty acids were methylated with KOH 0.2 M in anhydrous methanol and then analyzed via gas chromatography (GC 6890 N, Agilent Technologies, Santa Clara, CA, USA) using a 100 m × 0.25 mm × 0.20 µm fused silica capillary column (SP-2560, Supelco, Inc, Bellefonte, PA, USA). One microliter was injected into the chromatograph, which was equipped with a split/splitless injector and a flame ionization detector (FID). The oven temperature program was started at 150 °C, followed by increases of 1.50 °C/min up to 225 °C, at which point it was maintained for 15 min. The temperature of the injector and detector was 250 °C. The carrier gas was helium at 1 mL/min and the split ratio was 20:1. The different fatty acids were identified by the retention time using a mixture of fatty acid standards (47885-U Supelco, Sigma-Aldrich, Darmstadt, Germany). The fatty acid contents were calculated using chromatogram peak areas and were expressed as g per 100 g of total fatty acid methyl esters. All analyses were performed in triplicate.

### 2.9. Statistical Analysis

The significance of the effect was obtained via one-way analysis of variance (ANOVA) at an α = 0.05 level, using the F-test. The Tukey test was used to test for statistically significant differences between samples. All statistical analyses were carried out using SPSS Package 23 (IBM, Chicago, IL, USA).

## 3. Results and Discussion

The following sections show the results of the analysis of the different emulsions and products, with the exception of the frankfurter made with 100% texturized pea protein, owing to the poor textural characteristics of the final product.

### 3.1. Characterization of the Emulsion

The volume of total expressible fluid (TEF%) and the percentage of fat in this fluid, as well as the cooking loss, are important parameters in determining the characteristics of an emulsion, especially its ability to retain fat and moisture during processing [38]. Back-extrusion force is an important parameter since the emulsion has to be pumped through pipelines and frankfurter-forming equipment [39]. The results of these parameters are shown in Table 2. It is noteworthy that the results obtained for Meat (40%OO) were similar to those obtained by Lurueña-Martínez et al. [7] for the same formulation.

The results showed very good emulsion stability with values lower than 1% for most of the formulations. Indeed, a decrease in TEF%, although not significant (*p* > 0.05), was observed when all the backfat was replaced by olive oil ((100%OO) Meat) owing to the increase in locust bean/xantham gum in the formulation. This result is in agreement with those previously reported when a partial replacement was performed owing to the ability of this mixture to improve fluid retention even when the dose was increased by a small amount [7]. A similar trend was observed by other authors [40,41], who found a correlation between increased hydrocolloids in the formulation and reduced TEF%. On the other hand, as the percentage of lean substitution by vegetable texturized protein increased, the TEF% tended to increase and was statistically significant (*p* < 0.05) for the 75% formulation. This result differs from those of other works, which point to improved emulsion stability when vegetable protein (soy and gluten) were used as a total meat replacer [25].

The same behavior was observed for the fat % of the TEF, which showed a decrease between the 40% and 100%OO formulations owing to the higher addition of locust bean/xantham gum, as previously observed when the percentage of olive oil increased from zero to 40% [7]. Other studies have also shown a significant increase in emulsion stability when backfat was replaced by olive oil emulsions [17]. It was also observed that the fat % increased as the percentage of lean pork replaced with pea texturized protein increased, which is probably due to the progressive denaturation of the vegetable protein. However, the differences were not statistically significant (*p* > 0.05) and the values were always lower than those observed for Meat (40%OO), which confirms the strong capacity of this mixture for retaining monounsaturated fatty acids [7]. It is important to highlight that for frankfurters made with vegetable protein, most of the TEF released was water, in agreement with most previous studies [7,13,25,42].

Cooking loss showed the same trend as emulsion stability (TEF%) as previously reported [10,25], although more significant differences (*p* < 0.05) were observed for cooking loss. These differences are due to the fact that for the TEF % the temperature is reached very quickly and therefore gel formation and strength is improved in order to increase moisture retention [7]. A significant decrease (*p* < 0.05) was observed when fat was completely replaced by olive oil owing to the higher amount of hydrocolloids, as was previously reported for the processing yield when this mixture or lupin seeds were added [7,43]. On the other hand, other studies also showed a decrease in cooking losses as the percentage of the replacement of backfat by vegetable oils increased [10], which could be due to the higher fat content of Meat 100%OO (Table 3), as pointed out by Serdaroglu et al. [44]. In addition, the cooking loss decreased slightly when meat was partially substituted for vegetable protein at the amount of 25%, owing to its higher water-holding capacity [23,24,45]. However, as the percentage of substitution increases, the cooking loss also increases, and this was statistically significant (*p* < 0.05) for 75% substitution, in contrast to the results of [25]. This result could be due to differences in the type of interactions between polypeptide chains during heating, which leads to a higher release of water and fat.

Although olive oil is richer in unsaturated fatty acids, which could lead to a softer texture of the emulsion, as previously observed for pâté [46], no significant difference (*p* > 0.05) in back-extrusion force was observed when fat was fully replaced with olive oil. Furthermore, the substitution of pea protein for meat showed no significant difference (*p* > 0.05) in the back-extrusion force of up to 25% meat substitution. A significant decrease (*p* < 0.05) was, however, observed for 50% and 75% substitution. The use of legume flours as binders increases the viscosity of raw batters to reduce the problems related to fat reduction in cooked products [29]. To our knowledge, however, there are no data on the use of legume proteins as meat substitutes.

### 3.2. Proximate Composition

Results for the proximate composition of the frankfurters are shown in Table 3. The frankfurters obtained in this study were low-fat products because the final total fat content, which was between 13% and 16%, represents a reduction of around 35% compared to the usual fat content of these products, which is up to 30% [47]. On the other hand, there were significant differences (*p* < 0.05) between 40%OO and 100%OO Meat owing to the substitution of backfat for oil. This was due to the fact that backfat has a lower fat content (80–85%) than olive oil (100%), related to the presence of lean meat and inner membranes, as previously observed [7,46]. No significant changes (*p* > 0.05) were observed between 40%OO and 100%OO Meat for protein and moisture. The lower value observed for Meat 100%OO may be related to the higher fat content of this frankfurter. With regard to ash, the values of the Meat (100%OO) batches were significantly lower (*p* < 0.05) than those of the Meat (40%OO) batches. This result differs from those of other studies, which showed similar [13,46] or even higher values [48] for frankfurters with olive oil, although the differences were not statistically significant. However, Gao et al. [49] found a significant decrease when replacing animal fat with pre-emulsified sunflower oil.

The substitution of meat for texturized pea protein did not show significant differences (*p* > 0.05) in terms of total fat content from Meat formulations when taking intermediate values, although these were slightly lower than those of 100%OO owing to the lower fat content of texturized pea protein (3.1% vs. 4.8%). Moreover, the changes in the formulation did not affect the protein content (*p* > 0.05) owing mainly to the variability between batches, although a clear trend towards an increase in its levels was observed owing to the higher protein level of texturized pea protein (37.5% vs. 23.1%). Moisture content also showed a clear tendency towards a decrease as texturized pea protein was added to the frankfurter formulations. Significant differences (*p* < 0.05) were found between the batches with vegetable protein and the Meat 40%OO formulation and also between the batches with the higher percentage vegetable protein and the Meat 100%OO. This fact is mainly related with the lower moisture content of texturized pea protein and also with the higher cooking losses of TPP50% and especially of TPP75%.

On the other hand, the inclusion of texturized protein produced a slight tendency towards increased ash, which is in agreement with the studies of Colomer-Sellas [50] and is due to the higher percentage of ash in texturized pea protein (Table 3).

### 3.3. Instrumental Color and Texture

The color parameters (Table 4) showed a significant increase (*p* < 0.05) in the L* value and a significant decrease (*p* < 0.05) in a* when all the backfat was replaced with olive oil, but there were no significant differences for the b* parameter. These results differ from those obtained by other authors [46,48] in whose work the opposite trend was observed for the L* value, possibly attributable to the use of virgin olive oil which has a darker color (L* = 31.91) than olive oil (L* = 41.28). In contrast, in other studies which used olive oil the L* values were similar to or higher than those of the control [51]. In relation to the a* value, Álvarez et al. [52] found that the a* values in a meat emulsion made with olive oil were lower than in an emulsion made without olive oil. All the studies reflect a tendency to increase the b* value [46,47], although this increase is not always significant [53], as has been observed in this study. Some authors attribute this color alteration in olive oil sausages to the structural modifications which occur during the mincing process when the oily phase is distributed within the actomyosin matrix [48].

On the other hand, a progressive decrease in L* and a strong increase in b* with a higher percentage of replacement of meat by texturized pea protein was observed, whereas the a* value did not show a clear trend. This result was due to the more intense yellow color of texturized pea protein (b* = 35.99 vs. 13.13), whereas the a* values of lean pork and texturized pea protein was similar (9.20 vs. 10.12). A correlation between pulse flour color and the final product color was also observed by Canti et al. [45]. These results for b* and a* components are in agreement with those reported by Hidayat et al. [24] when using texturized vegetable protein to make sausages.

In terms of the L* value, textured pea protein has a slightly lighter color than lean pork (57.39 vs. 54.74); the progressive decrease in L* was therefore probably due to a reduction in meat pigments formed during the cooking process, together with a reduction in overall light scattering properties owing to differences in protein matrix structure [30]. This result is consistent with the progressive decrease in L* in the progressive substitution of meat for plant-based proteins [25] but differs from other results observed when meat was replaced with protein isolates [22,45] or texturized vegetable proteins in cooked products [24], for which either no significant differences were observed or there was a progressive increase in L*.

First of all, the texture analysis of texturized pea protein showed that the longitudinal and transversal forces were different (27.44 vs. 41.41 N) because the texturized fibers are arranged longitudinally in the extrusion direction. This makes it easier to cut in this direction and confirms that the product has a fibrous morphology, similar to that of cooked chicken meat [54].

Table 4 shows that the total replacement of pork backfat with olive oil did not affect (*p* > 0.05) any of the texture parameters analyzed. This can be explained by the higher content of locust bean gum/xanthan gum in the Meat (100%OO) formulation. In this sense Saldaña et al. [55] concluded that in the production of mortadella the effects of the replacement of animal fat with vegetable fat from cotton, sunflower, and palm oil were mitigated by the addition of hydrocolloids. This result reveals the strong influence of the addition of hydrocolloids on the properties of hardness, gumminess, and chewiness, with the use of these compounds being a good strategy for the substitution of animal fat for vegetable fat in terms of structure [27].

With regard to the progressive substitution of lean meat for texturized pea protein, all the parameters showed significant differences (*p* < 0.05) with the 100% meat-based formulations except adhesiveness. In all of them a progressive and significant decrease in values was observed as the substitution percentage increased. The texture values up to a percentage of 50% of meat substitution were within the range previously reported for frankfurters [7,23,25]. These results are in agreement with those obtained in cooked products when meat is replaced up to 40–60% by vegetable proteins, which showed a progressive and significant decrease in texture parameters [24,25]. However, the 75% replacement produced an excessive decrease in hardness, springiness, cohesiveness, gumminess, and chewiness, as observed by Kamani et al. [25] when the substitution increased from 60% to 100%.

Hardness and the related parameters (gumminess and chewiness) and cohesiveness, showed significant differences (*p* < 0.05) between 100% meat-based formulations and the frankfurters prepared with pea protein. This is probably due to the existence of a stronger network in myofibril proteins, which consequently increased the product’s resistance to compression [25]. However, as the percentage of texturized pea protein increased, these parameters showed a significant decrease. Previous works have shown that the addition of vegetable protein showed evidence regarding the interactions between vegetable and myofibrillar proteins that cause a less stable conformation [56] and interfere in the formation of a gel network in meat emulsions [57], which may result in a softer texture.

On the other hand, although the frankfurters made with 25% and 50% texturized pea protein showed a significantly lower value of springiness (*p* < 0.05) than 100% meat-based frankfurters, the values of these two formulations were similar to those of the Meat 40%OO and showed no significant differences (*p* > 0.05) between them. This result indicates that products containing up to 50% texturized pea protein showed good elasticity, probably due to the presence of hydrocolloids. However, TPP75% showed a significantly lower (*p* > 0.05) value of this parameter. The lower value of springiness may be due to the fact that non-meat proteins could hold more water and fat contents, which could fill the interstitial spaces within the protein matrix and reduce springiness [57].

### 3.4. Fatty Acid Composition

The fatty acid profiles of the elaborated products and of texturized pea protein are shown in Table 5. Although data on the fatty acid profile of pea protein isolates are very scarce, according to previous studies, the ranges in major fatty acid composition for whole peas are palmitic (C16:0) 10–16%, stearic (C18:0) 2.5–5.1%, oleic (C18:1) 22.6–36.0%, linoleic (C18:2 n6) 37.7–59.7%, and linolenic (C18:3 n3) 6.4–13.4% [58,59,60]. Therefore, the fatty acid composition of texturized pea protein used in this study was similar to those reported for whole pea bean. As for the ranges in minor fatty acid composition, the data obtained in this study were very similar to those reported by Padhi et al. [60] except for C14:0 (2.7% vs. 0.25%), whereas C16:1 (0.05–0.08% vs. 0.05%), C17:0 (0.14–0.18% vs. 0.22%), C20:0 (0.47–0.51% vs. 0.47%), C20:1 (0.03–0.04% vs. 0.02%), and C22:1 (0.03–0.07% vs. 0.01%) showed values similar to those reported in this study. Indeed, the total content of SFA, MUFA, and PUFA observed in texturized pea protein was in agreement with the values reported by these authors [60].

Therefore, texturized pea protein was characterized by its high content of C18:2 n6 compared with meat and olive oil, which is characterized by its high content of oleic acid which was higher than 70%, as previously reported [61,62]. As a result, the oleic acid content of the meat frankfurter (40%OO) was higher (*p* < 0.05) than the values found in all backfat low-fat pork sausages [53]. Furthermore, total fat replacement with olive oil produced a significant increase (*p* < 0.05) in C18:1, together with a significant decrease in C14:0, C16:1, C20:2 n6, C18:3 n6, C20:2 n6, C22:1 n9, and C24:1 n9 and a similar trend for C18:0 and C18:3n3, although in these cases the differences were not significant (*p* > 0.05). These results for C16:1, C18:1 n9, C14:0, and C20:2n6 fatty acids are in agreement with studies in which part or all of the fat in the meat product was replaced with olive oil [46,51] or all of the fat was replaced with an emulsion of a mixture of olive, linseed, and fish oils [63]. The different combination of oils would be responsible for the discrepancy in the results for the other fatty acids such as C18:3 n3 and C14:0. As for frankfurter-type sausages, studies by Domínguez et al. [53] confirm the significant decrease in C18:3 n3 and C20:3 n6, as well as C20:2 n6, with the replacement of fat with olive oil.

In relation to the sum of the different groups of fatty acids after the substitution of vegetable oil, a decrease in saturated fatty acids was observed, together with a significant increase (*p* < 0.05) in monounsaturated acids due to the richness of olive oil in oleic acid and a significant decrease (*p* < 0.05) in polyunsaturated acids. These results are due to the low presence of Omega 6 as C18:3 n6, C20:2 n6, and C18:2 n6 in olive oil, accounting for less than 7%, which also experienced a significant decrease. These results agree with what has previously been reported for various meat products [46,64,65]. As for the sum of n3 fatty acids, no significant variation was found in this study although they tended to decrease, as has already been observed by other authors [46,64].

The progressive replacement of meat by pea protein significantly increased (*p* < 0.05) the levels of C18:2 n6 for replacement levels of 50% and 75%. The increase was not as linear as expected, mainly due to the low fat retention of the emulsion observed in one of the trials made with 75% vegetable protein, which also caused a high standard deviation. A progressive and significant increase (*p* < 0.05) in the C18:3 n3 content was also observed. These results are correlated to the high proportion of these two fatty acids in texturized pea protein. A progressive decrease in C22:1 levels was also observed owing to the lower quantity of this acid in pea protein. Although progressive decreases in C14:0 and C17:1 were observed, the differences were not statistically significant (*p* > 0.05). As far as the remainder of major fatty acids is concerned there were no significant differences (*p* > 0.05) for C16:0, C18:0, or C18:1, which is mainly due to the low amount of fat in texturized pea protein. Owing to the aforementioned results, the substitution of pork meat for pea protein significantly increased (*p* < 0.05) PUFA contents owing to the significant increase in n6 and n3 PUFA in the higher substitution percentages, whereas SFA and MUFA remained stable and were not affected by the substitution of meat by texturizated pea protein.

The total replacement of backfat with olive oil did not significantly (*p* > 0.05) affect the polyunsaturated/saturated (P/S) fatty acid ratio, as previously reported [46,64]. However, replacing meat with 50% and 75% of texturized protein significantly increased (*p* < 0.05) this ratio. The P/S ratio of 100% meat and TPP25% products ranged from 0.51 to 0.55, which were similar to values reported for other meat products made with olive oil or highly oleic oils that ranged from 0.4 to 0.5 [46,64,66], whereas higher percentages of meat substitution (50% and 75%) exhibited values around 0.60. These values were slightly higher than those recommended (0.4) by the food standards agency COMA (Committee on Medical Aspects of Food Policy) [67]. A high PUFA proportion in itself is not necessarily healthy if it is not balanced in relation to the n6/n3 ratio, which should not be higher than four [68]. Therefore, the values observed for this ratio were higher than recommended. The replacement of backfat with olive oil significantly decreased the n6/n3 ratio, which differs from previously observations [46,64,66] but it is in agreement with the results of Pintado et al. [65]. On the other hand, the replacement of meat with texturized pea protein showed a progressive decrease in the n6/n3 ratio due to the high C18:3 n3 content of the pea protein.

## 4. Conclusions

The results obtained allow us to conclude that the complete substitution of animal fat for olive oil makes it possible to obtain a technologically viable product which only differs from the Meat 40%OO frankfurter in color (it is slightly lighter) and in its superior emulsion stability results. This result is attributable to the incorporation of hydrocolloids and specifically the locust bean/xantham gum mixture. Furthermore, this substitution improves the nutritional characteristics of frankfurters as they have a healthier lipid profile.

With regard to the incorporation of vegetable protein, it was observed that the products elaborated with a substitution of up to 50%, although they showed significant differences in the parameters of moisture, color, and texture, exhibited final values that were similar to those of other low-fat frankfurters. In addition, the fatty acid profile was healthier in relation to the Meat100%OO product, showing lower values of the n6/n3 ratio. It can therefore be concluded that textured pea protein can be a substitute for lean pork at up to 50% under the conditions tested. In view of the results obtained, it is recommended that future studies should modify the formulation with a higher inclusion of hydrocolloids to improve the final texture characteristics and achieve a higher substitution percentage.

## Figures and Tables

**Table 1 foods-11-00923-t001:** Low-fat frankfurter formulations with backfat replacement with olive oil and different levels of replacement of meat by texturized pea protein as a percentage of total weight.

Ingredients	Meat (40%OO)	Meat (100%OO)	TPP 25%	TPP50%	TPP75%	TPP100%
Lean pork	40	40	30	20	10	0
Texturized pea protein	0	0	10	20	30	40
Pork backfat	7.5	0	0	0	0	0
Olive oil	5	12.5	12.5	12.5	12.5	12.5
Locust bean/xantham gum	0.6	0.7	0.7	0.7	0.7	0.7
Ice	36.2	36.1	36.1	36.1	36.1	36.1
Polyphosphate	0.3	0.3	0.3	0.3	0.3	0.3
Nitrite salt ^1^	1.6	1.6	1.6	1.6	1.6	1.6
Potato starch	2.5	2.5	2.5	2.5	2.5	2.5
Soy protein	2	2	2	2	2	2
Sodium ascorbate	0.05	0.05	0.05	0.05	0.05	0.05
Dextrose	0.25	0.25	0.25	0.25	0.25	0.25
Sodium lactate	1	1	1	1	1	1
Flavorings	2	2	2	2	2	2
Walnut tree smoke	1	1	1	1	1	1
Onion	0.55	0.55	0.55	0.55	0.55	0.55
Garlic	0.4	0.4	0.4	0.4	0.4	0.4
Pepper	0.05	0.05	0.05	0.05	0.05	0.05

^1^ NaCl+0.6% sodium nitrite. TPP25%: substitution with 25% texturized pea protein; TPP50%: substitution with 50% texturized pea protein; TPP75%: substitution with 75% texturized pea protein.

**Table 2 foods-11-00923-t002:** Mean values (+SD) of emulsion stability, cooking loss, and emulsion texture for all frankfurter formulations.

	Meat (40%OO)	Meat (100%OO)	TPP25%	TPP50%	TPP75%
TEF%	0.93 ± 0.65 ^a^	0.17 ± 0.37 ^a^	0.33 ± 0.51 ^a^	0.45 ± 0.78 ^a^	3.60 ± 0.97 ^b^
EFat%	57.78 ± 32.79 ^b^	16.38 ± 34.52 ^a^	24.13 ± 32.67 ^a^	24.42 ± 35.63 ^a^	37.92 ± 40.20 ^a^
Cooking loss (%)	3.63 ± 0.32 ^b^	3.09 ± 0.14 ^a^	2.74 ± 0.14 ^a^	3.29 ± 0.21 ^a,b^	5.65 ± 0.47 ^c^
Back-extrusion force (N)	8.54 ± 0.69 ^a^	8.75 ± 0.39 ^a^	8.59 ± 2.02 ^a^	2.94 ± 1.08 ^b^	3.27 ± 0.02 ^b^

^a,b,c^ Different letters in the same row indicate statistically significant differences at *p* < 0.05. TPP25%: substitution with 25% texturized pea protein; TPP50%: substitution with 50% texturized pea protein; TPP75%: substitution with 75% texturized pea protein.

**Table 3 foods-11-00923-t003:** Mean values (+SD) of proximate composition parameters for raw materials and frankfurter formulations.

	Raw Ingredients		Frankfurter Formulations				
	Lean Pork	Texturized Pea Protein	Meat (40%OO)	Meat (100%OO)	TPP25%	TPP50%	TPP75%
Protein (%)	23.1 ± 1.7	37.5 ± 0.15	11.40 ± 0.92 ^a^	11.85 ± 0.57 ^a^	12.80 ± 0.57 ^a^	13.95 ± 0.49 ^a^	14.15 ± 2.05 ^a^
Total fat (%)	4.81 ± 1.5	3.10 ± 0.12	12.98 ± 1.14 ^a^	15.87 ± 1.43 ^b^	13.70 ± 2.27 ^a,b^	14.84 ± 2.22 ^a,b^	14.96 ± 0.63 ^a,b^
Moisture (%)	71.1 ± 1.1	56.10 ± 0.11	66.69 ± 4.33 ^c^	63.45 ± 2.69 ^b,c^	61.39 ± 0.23 ^a,b^	58.66 ± 0.12 ^a^	58.98 ± 1.53 ^a^
Ash (%)	0.98 ± 0.2	2.9 ± 0.05	2.20 ± 0.74 ^b^	0.98 ± 0.24 ^a^	1.14 ± 0.33 ^a^	1.25 ± 0.21 ^a^	1.35 ± 0.46 ^a^

^a,b,c^ Different letters in the same row indicate statistically significant differences at *p* < 0.05. TPP25%: substitution with 25% texturized pea protein; TPP50%: substitution with 50% texturized pea protein; TPP75%: substitution with 75% texturized pea protein.

**Table 4 foods-11-00923-t004:** Mean values (+SD) of color and texture parameters for all frankfurter formulations.

	Meat (40%OO)	Meat (100%OO)	TPP25%	TPP50%	TPP75%
L*	69.61 ± 3.79 ^a,b^	74.98 ± 4.06 ^c^	71.71 ± 1.38 ^b,c^	72.07 ± 1.69 ^b,c^	68.24 ± 1.31 ^a^
a*	8.17 ± 1.04 ^b^	6.35 ± 1.16 ^a^	7.42 ± 0.20 ^b^	6.25 ± 0.15 ^a^	7.29 ± 1.00 ^b^
b*	15.34 ± 2.34 ^a^	16.78 ± 2.43 ^a,b^	19.25 ± 1.92 ^b,c^	21.29 ± 2.16 ^c^	24.76 ± 2.90 ^d^
Hardness (gf)	3076.26 ± 440.78 ^d^	3020.56 ± 411.96 ^d^	2601.18 ± 591.22 ^c^	1931.62 ± 511.09 ^b^	546.40 ± 148.93 ^a^
Adhesiveness (gf mm)	−26.62 ± 14.37 ^a^	−31.12 ± 11.66 ^a^	−37.34 ± 21.00 ^a^	−36.02 ± 16.74 ^a^	−31.95 ± 22.80 ^a^
Springiness (mm)	0.89 ± 0.37 ^c^	0.91 ± 0.31 ^c^	0.84 ± 0.34 ^b^	0.83 ± 0.29 ^b^	0.67 ± 0.82 ^a^
Cohesiveness	0.78 ± 0.31 ^d^	0.78 ± 0.17 ^d^	0.70 ± 0.59 ^c^	0.60 ± 0.60 ^b^	0.35 ± 0.42 ^a^
Gumminess (gf)	2382.58 ± 320.05 ^d^	2356.13 ± 294.51 ^d^	1843.26 ± 480.27 ^c^	1154.26 ± 255.24 ^b^	187.31 ± 46.62 ^a^
Chewiness (gf mm)	2114.04 ± 237.44 ^d^	2148.23 ± 241.93 ^d^	1556.88 ± 437.65 ^c^	957.74 ± 217.25 ^b^	125.46 ± 30.69 ^a^

^a,b,c,d^ Different letters in the same row indicate statistically significant differences at *p* < 0.05. L*: Lightness; a*: redness, b*: yellowness.

**Table 5 foods-11-00923-t005:** Mean values (+SD) of fatty acid composition for all frankfurter formulations.

	Texturized Pea Protein	Meat (40%OO)	Meat (100%OO)	TPP25%	TPP50%	TPP75%
C12:0	n.d.	0.03 ± 0.02 ^a^	0.02 ± 0.01 ^a^	0.02 ± 0.02 ^a^	0.01 ± 0.02 ^a^	0.01 ± 0.01 ^a^
C14:0	0.25 ± 0	0.58 ± 0.10 ^b^	0.32 ± 0.21 ^a^	0.26 ± 0.14 ^a^	0.28 ± 0.09 ^a^	0.20 ± 0.08 ^a^
C14:1	n.d.	n.d. ^a^	n.d.^a^	0.001 ± 0 ^a^	0.006 ± 0.01 ^a^	n.d. ^a^
C15:0	n.d.	0.02 ± 0.02 ^a^	0.02 ± 0.01 ^a^	0.01 ± 0.01 ^a^	0.02 ± 0.01 ^a^	0.02 ± 0.01 ^a^
C16:0	13.31 ± 0.24	15.61 ± 0.53 ^b^	12.98 ± 0.79 ^a^	12.78 ± 2.00 ^a^	13.50 ± 1.07 ^a^	12.45 ± 1.34 ^a^
C16:1 n7	0.03 ± 0.01	0.17 ± 0.15 ^a^	0.17 ± 0.03 ^a^	0.16 ± 0.02 ^a^	0.29 ± 0.36 ^a^	0.15 ± 0 ^a^
C16:1	0.05 ± 0	1.57 ± 0.09 ^b^	1.06 ± 0.07 ^a^	1.04 ± 0.28 ^a^	1.11 ± 0.10 ^a^	0.95 ± 0.15 ^a^
C17:0	0.22 ± 0.01	0.18 ± 0.01 ^a^	0.10 ± 0.03 ^a^	0.13 ± 0.01 ^a^	0.11 ± 0.05 ^a^	0.11 ± 0.02 ^a^
C17:1	0.06 ± 0.01	0.18 ± 0.02 ^a^	0.15 ± 0 ^a^	0.11 ± 0.07 ^a^	0.17 ± 0.01 ^a^	0.14 ± 0.03 ^a^
C18:0	4.31 ± 0.14	7.33 ± 1.58 ^a^	5.48 ± 1.34 ^a^	5.57 ± 0.65 ^a^	5.27 ± 1.19 ^a^	4.85 ± 0.54 ^a^
C18:1	28.45 ± 0.09	58.74 ± 3.44 ^a^	68.12 ± 2.72 ^b^	67.74 ± 2.63 ^b^	65.81 ± 1.43 ^b^	68.83 ± 4.14 ^b^
C18:1 n7	n.d.	1.10 ± 1.82 ^a^	0.97 ± 1.11 ^a^	1.15 ± 1.28 ^a^	0.49 ± 0.90 ^a^	0.02 ± 0.02 ^a^
C18:2 n6	44.96 ± 0.69	11.02 ± 1.07 ^b^	7.94 ± 1.08 ^a^	7.72 ± 1.79 ^a^	9.93 ± 0.15 ^a,b^	9.26 ± 2.06 ^a,b^
C20:0	0.47 ± 0.02	0.35 ± 0 ^a^	0.40 ± 0.03 ^a,b^	0.49 ± 0.10 ^b^	0.42 ± 0.01 ^a,b^	0.43 ± 0.01 ^a,b^
C18:3 n3	6.98 ± 0.20	0.73 ± 0.07 ^b^	0.54 ± 0.37 ^a^	0.86 ± 0.02 ^a,b,c^	1.06 ± 0.14 ^b,c^	1.16 ± 0.03 ^c^
C18:3 n6	0.01 ± 0.01	0.67 ± 0.21 ^b^	0.48 ± 0.21 ^a,b^	0.31 ± 0.19 ^a^	0.40 ± 0.03 ^a,b^	0.36 ± 0.02 ^a,b^
C20:1	0.02 ± 0.00	0.02 ± 0.02 ^a^	0.01 ± 0.01 ^a^	0.01 ± 0.01 ^a^	0.01 ± 0.01 ^a^	0.01 ± 0.01 ^a^
C20:2 n6	0.06 ± 0.00	0.34 ± 0.04 ^b^	0.12 ± 0.05 ^a^	0.07 ± 0.05 ^a^	0.10 ± 0.02 ^a^	0.07 ± 0.01 ^a^
C22:0	n.d.	0.19 ± 0.02 ^a^	0.15 ± 0.01 ^a^	0.19 ± 0.05 ^a^	0.15 ± 0.01 ^a^	0.14 ± 0.01 ^a^
C20:3 n6	0.05 ± 0.00	0.07 ± 0.00 ^b^	0.03 ± 0.01 ^a^	0.01 ± 0.01 ^a^	0.01 ± 0.01 ^a^	0.01 ± 0.01 ^a^
C22:1	0.01 ± 0.01	0.36 ± 0.02 ^e^	0.21 ± 0.01 ^d^	0.16 ± 0.02 ^c^	0.12 ± 0.01 ^b^	0.09 ± 0.00 ^a^
C20:3 n3	0.04 ± 0.05	0.01 ± 0.01 ^a^	0.03 ± 0.02 ^a^	0.01 ± 0.01 ^a^	0.02 ± 0.01 ^a^	0.03 ± 0.00 ^a^
C20:4 n6	n.d	0.38 ± 0.18 ^a^	0.51 ± 0.16 ^a^	1.05 ± 0.87 ^a^	0.37 ± 0.31 ^a^	0.55 ± 0.26 ^a^
C23:0	n.d	n.d. ^a^	n.d. ^a^	0.01 ± 0.01 ^a^	0.19 ± 0.36 ^a^	0.01 ± 0.00 ^a^
C22:2 n6	0.16 ± 0.23	0.05 ± 0.01 ^a^	0.06 ± 0.01 ^a^	0.09 ± 0.03 ^a^	0.07 ± 0.01 ^a^	0.08 ± 0.01 ^a^
C24:1	0.12 ± 0.16	0.11 ± 0.01 ^c^	0.05 ± 0.01 ^b^	0.02 ± 0.02 ^a^	0.03 ± 0.01 ^a,b^	0.03 ± 0.01 ^a,b^
C20:5 n3	0.09 ± 0.01	0.01 ± 0.01 ^a^	0.01 ± 0.01 ^a^	0.01 ± 0.02 ^a^	0.01 ± 0.01 ^a^	0.01 ± 0.01 ^a^
SFA	18.57 ± 0.41	24.29 ± 2.20 ^b^	19.51 ± 2.36 ^a,b^	19.46 ± 2.68 ^a,b^	19.95 ± 2.17 ^a,b^	18.23 ± 2.02 ^a^
MUFA	28.87 ± 0.20	62.41 ± 1.47 ^a^	70.75 ± 1.51 ^b^	70.40 ± 3.51 ^b^	68.04 ± 1.90 ^b^	70.22 ± 1.60 ^b^
PUFA	52.56 ± 0.06	13.29 ± 0.73 ^c^	9.74 ± 0.89 ^a^	10.14 ± 0.85 ^a,b^	12.00 ± 0.35 ^b^	11.55 ± 0.90 ^b^
n3	0.12 ± 0.01	0.75 ± 0.9 ^a^	0.71 ± 0.10 ^a^	0.77 ± 0.20 ^a^	1.09 ± 0.13 ^b^	1.20 ± 0.04 ^b^
n6	52.16 ± 0.06	12.51 ± 0.91 ^b^	9.09 ± 1.11 ^a^	9.80 ± 0.93 ^a^	11.53 ± 0.39 ^a,b^	11.13 ± 0.85 ^a,b^
PUFA/SFA		0.55 ± 0.03 ^a^	0.51 ± 0.05 ^a^	0.52 ± 0.03 ^a^	0.60 ± 0.05 ^b^	0.63 ± 0.03 ^b^
n6/n3		16.62 ± 0.92 ^c^	12.86 ± 1.71 ^b^	11.42 ± 1.22 ^a,b^	10.68 ± 1.05 ^a,b^	9.23 ± 0.26 ^a^

^a,b,c,d^ Different letters in the same row indicate statistically significant differences at *p* < 0.05. n.d. not detected.

## Data Availability

Not applicable.

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
