# Peer review of "The Effects of the Progressive Replacement of Meat with Texturized Pea Protein in Low-Fat Frankfurters Made with Olive Oil"

_foods, 2022, doi:10.3390/foods11070923_

Round 1

Reviewer 1 Report

The manuscript is well written however there are few comments as per indicated in the attached file for consideration. 

Author Response

All the responses to the reviewer suggestions have been included in the pdf document and the changes highlighted in red in the revised version

Reviewer 2 Report

Dear Editor,

This work was designed to understand the effect total substitution of animal fat for olive oil in the production of low-fat frankfurters and then to proceed to the progressive replacement of lean meat with texturized pea protein on stability and quality of frankfurters. Vegan/vegetarian approach of authors increase the value of study since food industry is moving towards to vegan/vegetarian approach.

However, I have some concerns about the manuscript such as;

  • I think selection of control group and title of manuscript does not match and using one-way Anova method could not be enough for explaining effects of different fat replacement, gum and pea protein ratio. I recommend authors could think changing design of manuscript and extract 40%OO group.
  • I have some concerns about the results of some analysis written below.
  • Authors should add some indexes or maybe cholesterol content to identify sausages as “healthier”.

The manuscript could be accepted after considering major revisions as shown below:

  1. Line 53: please remove “.” between pre-emulsified oil. or liquid oil
  2. Line 96: please specify the part of pork.
  3. Line 97: please give the fatty acid profile of olive oil
  4. Line 115: I think authors should indicate that texturized pea protein (TPP) groups were elaborated by changes in 100%OO groups. So, when the authors said Control group is 40%OO, it does not seem right comparing TPP groups with Control group since two variables affect the results: pork back fat ratio and TPP.
  5. Table 1.: please change the codes of TPP groups such as TPP25, TPP50 etc.
  6. Line 122: please define the low speed and high speed (rpm?)
  7. Line 142: please give the reference for this method.
  8. Line 149: fat percentage does not seem right; expressible fat is more accurate.
  9. Line 206 statistical analysis: I recommend authors to use two-way Anova to compare results of samples since comparing control (40%OO) to TPP groups is not valid to understand effects fat replacement, gum percentage, or TPP addition.
  10. All the Tables, please add row or column to the footnotes. “Different letters in same row mean statistically significant differences at p<0.05”
  11. All the results; please add p<0.05 (for significant), p>0.05 (not significant) in the sentences.
  12. Table 2, TEF% - Fat% : I recommend authors to repeat these analysis since standard deviation is too much even higher than means of samples. This results, surely, does not show significant differences.
  13. Table 2, TEF% and Cooking Loss%: Comparing to 100%OO and %25 samples, TEF% is higher but cooking loss is lower in %25. Authors should explain the reason of this behavior.
  14. Line 252: Please check the layout of references; [31,44] instead of (Hughes and Kamani).
  15. Moisture% results: I agree that high protein% and water retention results could increase the moisture% of samples. However, 75% sample had the highest TPP ratio (41% moisture), highest TEF% and cooking loss% but had the highest moisture%. Authors should check the results and explain this behavior.
  16. Table 4.: Please check the a* result of 50%.
  17. Line 339: Kamani et al [31]
  18. Table 4.: Please add units of texture results.
  19. Line 357: Please rewrite the sentence: With regard to the progressive substitution of lean meat for texturized pea protein, all the parameters showed significant differences with the 100% meat-based formulations except adhesiveness.
  20. Texture results: please improve the discussion of the part.
  21. Line 429: To say “healthier” for a product authors should calculate indexes such as atherogenic/thrombogenic and/or n3/n6 etc. then compare with the other studies or reports.

Author Response

The authors are grateful for the reviewer's comments. Changes made according to the reviewer's suggestions are highlighted in blue in the text.

Dear Editor,

This work was designed to understand the effect total substitution of animal fat for olive oil in the production of low-fat frankfurters and then to proceed to the progressive replacement of lean meat with texturized pea protein on stability and quality of frankfurters. Vegan/vegetarian approach of authors increase the value of study since food industry is moving towards to vegan/vegetarian approach.

However, I have some concerns about the manuscript such as;

  • I think selection of control group and title of manuscript does not match and using one-way Anova method could not be enough for explaining effects of different fat replacement, gum and pea protein ratio. I recommend authors could think changing design of manuscript and extract 40%OO group.
  • I have some concerns about the results of some analysis written below.
  • Authors should add some indexes or maybe cholesterol content to identify sausages as “healthier”.

The manuscript could be accepted after considering major revisions as shown below:

  1. Line 53: please remove “.” between pre-emulsified oil. or liquid oil

It has been done

  1. Line 96: please specify the part of pork.

The part of the pork was shoulder. It has been added.

  1. Line 97: please give the fatty acid profile of olive oil

It has been added.

  1. Line 115: I think authors should indicate that texturized pea protein (TPP) groups were elaborated by changes in 100%OO groups. So, when the authors said Control group is 40%OO, it does not seem right comparing TPP groups with Control group since two variables affect the results: pork back fat ratio and TPP.

We have changed control group by Meat40%OO group and we have specified that TPP groups are changes in 100%OO group.

  1. Table 1.: please change the codes of TPP groups such as TPP25, TPP50 etc.

It has been changed.

  1. Line 122: please define the low speed and high speed (rpm?)

Low speed: 1500 rpm, high speed 3000 rpm

  1. Line 142: please give the reference for this method.

It has been added

  1. Line 149: fat percentage does not seem right; expressible fat is more accurate.

The reference method uses name to refer to this parameter. Therefore, we consider more accurate to keep it

  1. Line 206 statistical analysis: I recommend authors to use two-way Anova to compare results of samples since comparing control (40%OO) to TPP groups is not valid to understand effects fat replacement, gum percentage, or TPP addition.

After having consulted an expert in statistical analysis of the data and taking into account that the other reviewers have not requested to change the analysis performed, we have decided to keep the ANOVA analysis. Although it is true that there are two factors considered, the design has been done in batches, the change has been done sequentially and the interpretation allows us to establish which changes are due to the oil and which are due to the vegetable protein. In any case, we are open to discuss the issue with the reviewer.

  1. All the Tables, please add row or column to the footnotes. “Different letters in same row mean statistically significant differences at p<0.05”

It has been added

  1. All the results; please add p<0.05 (for significant), p>0.05 (not significant) in the sentences.

It has been added

  1. Table 2, TEF% - Fat% : I recommend authors to repeat these analysis since standard deviation is too much even higher than means of samples. This results, surely, does not show significant differences.

The was a mistake in the SD of TEF% and it has been changed.

It is true that the deviation of Fat% is very high, but this is because the two trials showed very different values of this parameter. We repeated this several times but we did not manage to reduce the SD value.

  1. Table 2, TEF% and Cooking Loss%: Comparing to 100%OO and %25 samples, TEF% is higher but cooking loss is lower in %25. Authors should explain the reason of this behavior.

It is true that the TEF% value is higher and the cooking loss value is lower for TPP25% but the differences were not statistically significant. Therefore, considering the small numerical difference, we have not included an explanation for this fact.

  1. Line 252: Please check the layout of references; [31,44] instead of (Hughes and Kamani).

It has been changed

  1. Moisture% results: I agree that high protein% and water retention results could increase the moisture% of samples. However, 75% sample had the highest TPP ratio (41% moisture), highest TEF% and cooking loss% but had the highest moisture%. Authors should check the results and explain this behavior.
  2. Table 4.: Please check the a* result of 50%.

We have reviewed the data and made again the statistical analysis and this data is correct although unexpected.

  1. Line 339: Kamani et al [31]

It has been changed

  1. Table 4.: Please add units of texture results.

They have been added

  1. Line 357: Please rewrite the sentence: With regard to the progressive substitution of lean meat for texturized pea protein, all the parameters showed significant differences with the 100% meat-based formulations except adhesiveness.

It has been changed

  1. Texture results: please improve the discussion of the part.

We have tried to improve this section including more detailed explanations.

  1. Line 429: To say “healthier” for a product authors should calculate indexes such as atherogenic/thrombogenic and/or n3/n6 etc. then compare with the other studies or reports.

The n3/n6 and PUFA/SFA ratios has been calculated and a comparison with other studies has been included

Reviewer 3 Report

The following points must be clarified:

Statistical analysis: Randomized complete block design or completely randomized design?

Line 139:  It is thought that polyethylene is not appropriate for vacuum packing. Technical properties of this material must be given.
Line 212-214: Results for the frankfurters produced with 100% texturized pea protein must be presented. By this way the readers will have the opportunity to make comparisons.  
Line 223:  Results regarding the formulation that contains 100% texturized pea protein must be given in Table 2 as well as in other tables.
Line 287: Percentage humidity values given in Table 3 for Frankfurter formulations are quite low (33.31±4.33a, 36.55±2.69a,b 38.60±0.23b,c 41.34±0.12c 41.33±3.82c). This product is an emulsified meat product and its production contains addition of ice, as the author(s) did. How can you explain these values?
- The difference between Meat (40%OO) and Meat (100%OO) groups in  Table 3 must be explained.  
-Table 5: The results for 18:2n6 is quite controversial. Please re-check.

Author Response

The authors are grateful for the reviewer's comments. Changes made according to the reviewer's suggestions are highlighted in green in the text.

Statistical analysis: Randomized complete block design or completely randomized design?

Completely randomized design

Line 139:  It is thought that polyethylene is not appropriate for vacuum packing. Technical properties of this material must be given.

The pouches used were those recommended by the vacuum packaging machine company. (Tecnotrip) for cooking/pasteurisation. The outside is made of polyamide and the inside of polyethylene for thermal sealing. The information has been added to the text.

Line 212-214: Results for the frankfurters produced with 100% texturized pea protein must be presented. By this way the readers will have the opportunity to make comparisons.

Line 223:  Results regarding the formulation that contains 100% texturized pea protein must be given in Table 2 as well as in other tables.

We are very sorry but we did not analyse this trial because the texture was so deficient that it was impossible to even peel the frankfurters and we discarded it before to start the analysis.

Line 287: Percentage humidity values given in Table 3 for Frankfurter formulations are quite low (33.31±4.33a, 36.55±2.69a,b 38.60±0.23b,c 41.34±0.12c 41.33±3.82c). This product is an emulsified meat product and its production contains addition of ice, as the author(s) did. How can you explain these values?

The reviewer is right the results are not correct because it corresponded to dry matter instead of moisture. This error has been corrected in the new version and all the discussion changed according to the new data.

- The difference between Meat (40%OO) and Meat (100%OO) groups in  Table 3 must be explained.

This part has been changed in order to better highlight the differences between Meat (40%OO) and Meat (100%OO).

-Table 5: The results for 18:2n6 is quite controversial. Please re-check

We believe that the results are not controversial. It is true that the increase is not as linear as expected mainly due to the low fat retention of the emulsion observed in one of the trials made with 75% vegetable protein. This explanation has been added to the text.

Reviewer 4 Report

General comment:

The paper has interesting data obtained through the use of well-selected methods and analyses. The paper provides data that will complement those from previously published research regarding emulsified-type meat products with improved nutritional properties.

However,  the authors should make some improvements, namely in the Results and Discussion section.

OBJECTIONS

Abstract

  • Lines 16–19: Please rewrite this sentence according to comments.

Introduction

The Introduction is very well written and the authors clearly explained the reason for the use of olive oil and texturized plant proteins in emulsified-type meat products. The aim of the research is also clearly defined.

Some minor corrections could improve this section:

  • Line 78, page 2: “using soy proteins”
  • Line 85, page 2: Please provide the correct number for citations [36,36].
  • Line 93, page 2: Correct “products manufactured” into “frankfurters”.

Materials and methods

Methods are well chosen and could provide sufficient data for discussion and conclusions.

Some minor corrections could improve this section:

  • Line 111, page 3: Please explain the reason you chose treatment Meat40%OO as control? Why didn’t you choose treatment with all backfat?

RESULTS AND DISCUSSION

The presentation of results and discussion is at some points inadequate, therefore some sentences should be rephrased.

  • Lines 235–238, page 6: The absolute values are not comparable due to many differences of the products being compared: formulation (% of meat and fat used), meat type (chicken or pork), oil pre-treatment (liquid or emulsified), amount of non-meat ingredients used etc. It is more appropriate to compare trends. This part should be rewritten.
  • Line 252, page 6: References should be written with numbers in brackets.
  • Lines 258–260, page 6: There is also an inadequate citation here. In papers 49–51 control treatments are with all animal fat (pork or beef) while in this research the control is with olive oil and backfat. So it is more adequate to compare the effect of the fat substitution level on examined properties. Moreover, reference [51] examines meat products without any addition of oil. Also Serdaroglu et al. is under number 51 in References.
  • Line 262, page 7: add in amount of 25% between protein and owing
  • Line 278, page 7: add obtained in this research after Frankfurters.
  • Lines 280-283, page 7: This is a very confusing explanation. The differences in fat content between Meat(40%OO) and Meat(100%OO) were due to differences in total fat content of backfat (usually 80–85%) and olive oil (100%). Moreover, reference [42] is about the composition of lamb subcutaneous fat!
  • Line 291, page 7: add protein between higher and level
  • Line 295–299, page 7: First, moisture content is very low regarding frankfurters. The sum of protein, total fat, moisture and ash contents is about 60% in control (similarly to other treatments). Please explain this. Moreover, more clear explanation is needed to explain the increase of moisture content in frankfurters with a higher level of texturized pea protein (TPP) despite its lower moisture content compared to lean pork. Ice amount was similar in all treatments, while coking loss was similar or higher in frankfurters with TPP. Where does that moisture come from?
  • Lines 300–301, page 7: Please rewrite this, it is unclear.
  • Line 308, page 8: The author’s name must be in front of [56].
  • Line 315, page 8: Data about colour properties should be provided for the following statements: “…possibly attributable to the use of virgin olive oil which has a darker color” and “…in other studies which did not use this type of olive oil”.
  • Line 339, page 8: Reference Kamani et al. should be written with the number in brackets.
  • Lines 348–348, page 9: Statement „…total replacement of pork backfat with olive oil did not affect any of the texture parameters analyzed which differs from the results obtained by Lurueña-Martı́nez et al. [8]. …“ has to be rewritten, because it is correct only when compared to the treatment with backfat partly substituted with olive oil (there is no treatment with all backfat). Since the control treatment is not with all backfat, it cannot be concluded that there is no effect on texture parameters. Moreover, Lurueña-Martı́nez et al. in their research did not totally replace backfat with olive oil.
  • Lines 361–362, page 9: This statement should be deleted because acceptable values for texture properties have not been established! Do not compare the absolute values of instrumental texture properties (as well as colour), compare trends.
  • Lines 415–418, page 11: This section should be rewritten:
    • “...while n3 and SFA tended to decrease although the differences were not statistically significant as in the MUFA” – values of n-3 and SFA were lower in treatments with TPP, however no decreasing trend was observed with higher TPP level; MUFA remains stable in all treatments with TPP compared to 100%OO. The significant increase of MUFA in 100%OO compared to 40%OO is the effect of the higher level of olive oil, not TPP addition.
  • Line 422, page 11: It is necessary to emphasize the formulation of the control treatment.
  • Lines 426–431: Without sensory analysis it cannot be concluded that the obtained instrumental properties are acceptable. Rewrite this section.

Author Response

General comment:

The paper has interesting data obtained through the use of well-selected methods and analyses. The paper provides data that will complement those from previously published research regarding emulsified-type meat products with improved nutritional properties.

However,  the authors should make some improvements, namely in the Results and Discussion section.

OBJECTIONS

The authors are grateful for the reviewer's comments. Changes made according to the reviewer's suggestions are highlighted in purple in the text.

Abstract

  • Lines 16–19: Please rewrite this sentence according to comments.

It has been rewritten

Introduction

The Introduction is very well written and the authors clearly explained the reason for the use of olive oil and texturized plant proteins in emulsified-type meat products. The aim of the research is also clearly defined.

Some minor corrections could improve this section:

  • Line 78, page 2: “using soy proteins”

It has been added

  • Line 85, page 2: Please provide the correct number for citations [36,36].

It has been corrected

  • Line 93, page 2: Correct “products manufactured” into “frankfurters”.

Materials and methods

Methods are well chosen and could provide sufficient data for discussion and conclusions.

Some minor corrections could improve this section:

  • Line 111, page 3: Please explain the reason you chose treatment Meat40%OO as control? Why didn’t you choose treatment with all backfat?

Firstly, taking into account the suggestion of another reviewer we have changed the control to Meat40%OO throughout the text.

Secondly, we have decided not to elaborate all backfat product because we compare this type of frankfurter with Meat40%OO several years ago (Lurueña et al., 2004) and the differences among them were very small. We have repeated both frankfurters during the last years obtaining very similar results in terms of emulsion stability, texture and sensory acceptability (unpublished results). This fact is reflected in the following sentence in the text “This formulation shows no significant differences from a low-fat frankfurter elaborated only with backfat regarding emulsion stability, texture, or consumer acceptability [8].”

The number of different frankfurters should have been 6, which is a significant number of products. Therefore, as we have characterised both frankfurters very well, we decided to start with a new trial using Meat40%OO as a starting point.

RESULTS AND DISCUSSION

The presentation of results and discussion is at some points inadequate, therefore some sentences should be rephrased.

  • Lines 235–238, page 6: The absolute values are not comparable due to many differences of the products being compared: formulation (% of meat and fat used), meat type (chicken or pork), oil pre-treatment (liquid or emulsified), amount of non-meat ingredients used etc. It is more appropriate to compare trends. This part should be rewritten.

We have deleted the comparison between our results and the previous ones, and have only left the comparison on the emulsion stability behaviour when vegetable proteins are added.

  • Line 252, page 6: References should be written with numbers in brackets.

It has been changed

  • Lines 258–260, page 6: There is also an inadequate citation here. In papers 49–51 control treatments are with all animal fat (pork or beef) while in this research the control is with olive oil and backfat. So it is more adequate to compare the effect of the fat substitution level on examined properties. Moreover, reference [51] examines meat products without any addition of oil. Also Serdaroglu et al. is under number 51 in References.

We have revised this part. The reference numbers have been completely changed, but there was an error in the numbering in the previous version, because of this the reviewer did not find a correlation between them and the explanation.

We have included some references regarding the effect of animal fat substitution level by vegetable oils (we did not find any reference with olive oil for this parameter).

We use Serdaroglu's reference to show that as fat content decreases, cooking losses increase.

  • Line 262, page 7: add in amount of 25% between protein and owing

It has been added

  • Line 278, page 7: add obtained in this researchafter Frankfurters.

It has been added

  • Lines 280-283, page 7: This is a very confusing explanation. The differences in fat content between Meat(40%OO) and Meat(100%OO) were due to differences in total fat content of backfat (usually 80–85%) and olive oil (100%). Moreover, reference [42] is about the composition of lamb subcutaneous fat!

As noted above, there was an error in the numbering in the previous version, because of this the reviewer did not find a correlation between them and the explanation.

We have rewritten this part in order to clarify the explanation.

  • Line 291, page 7: add proteinbetween higher and level

It has been added

  • Line 295–299, page 7: First, moisture content is very low regarding frankfurters. The sum of protein, total fat, moisture and ash contents is about 60% in control (similarly to other treatments). Please explain this. Moreover, more clear explanation is needed to explain the increase of moisture content in frankfurters with a higher level of texturized pea protein (TPP) despite its lower moisture content compared to lean pork. Ice amount was similar in all treatments, while coking loss was similar or higher in frankfurters with TPP. Where does that moisture come from?
  • Lines 300–301, page 7: Please rewrite this, it is unclear.

The reviewer is right the results are not correct because it corresponded to dry matter instead of moisture. This error has been corrected in the new version and all the discussion changed according to the new data.

  • Line 308, page 8: The author’s name must be in front of [56].

It has been added

  • Line 315, page 8: Data about colour properties should be provided for the following statements: “…possibly attributable to the use of virgin olive oil which has a darker color” and “…in other studies which did not use this type of olive oil”.

The L* values for extra virgin olive oil and olive oil has been added.

  • Line 339, page 8: Reference Kamani et al. should be written with the number in brackets.

It has been changed

  • Lines 348–348, page 9: Statement „…total replacement of pork backfat with olive oil did not affect any of the texture parameters analyzed which differs from the results obtained by Lurueña-Martı́nez et al. [8]. …“ has to be rewritten, because it is correct only when compared to the treatment with backfat partly substituted with olive oil (there is no treatment with all backfat). Since the control treatment is not with all backfat, it cannot be concluded that there is no effect on texture parameters. Moreover, Lurueña-Martı́nez et al. in their research did not totally replace backfat with olive oil.

It has been changed

  • Lines 361–362, page 9: This statement should be deleted because acceptable values for texture properties have not been established! Do not compare the absolute values of instrumental texture properties (as well as colour), compare trends.

The sentence has been changed to show that the texture values were within the range described for other frankfurters.

  • Lines 415–418, page 11: This section should be rewritten: “...while n3 and SFA tended to decrease although the differences were not statistically significant as in the MUFA” – values of n-3 and SFA were lower in treatments with TPP, however no decreasing trend was observed with higher TPP level; MUFA remains stable in all treatments with TPP compared to 100%OO. The significant increase of MUFA in 100%OO compared to 40%OO is the effect of the higher level of olive oil, not TPP addition.

The section has been rewritten, according to reviewer suggestion and taking into account that the revision of the data showed that there was a mistake in the calculation of the total n3.

  • Line 422, page 11: It is necessary to emphasize the formulation of the control treatment.

It has been changed

  • Lines 426–431: Without sensory analysis it cannot be concluded that the obtained instrumental properties are acceptable. Rewrite this section.

It has been rewritten

Round 2

Reviewer 3 Report

Accept in present form